# Zebrafish and Flavonoids: Adjuvants against Obesity

**DOI:** 10.3390/molecules26103014

**Published:** 2021-05-19

**Authors:** Giuseppe Montalbano, Kamel Mhalhel, Marilena Briglia, Maria Levanti, Francesco Abbate, Maria Cristina Guerrera, Enrico D’Alessandro, Rosaria Laurà, Antonino Germanà

**Affiliations:** 1Zebrafish Neuromorphology Lab, Department of Veterinary Sciences, Via Palatucci snc, University of Messina, 98168 Messina, Italy; kamel.mhalhel@unime.it (K.M.); marilena.briglia@unime.it (M.B.); mblevanti@unime.it (M.L.); abbatef@unime.it (F.A.); mcguerrera@unime.it (M.C.G.); laurar@unime.it (R.L.); agermana@unime.it (A.G.); 2Unit of Animal Production, Department of Veterinary Sciences, Via Palatucci snc, Messina University, 98168 Messina, Italy; enrico.dalessandro@unime.it

**Keywords:** zebrafish, flavonoids, polyphenols, obesity, inflammation, obesity models, natural compounds, metabolic syndrome

## Abstract

Obesity is a pathological condition, defined as an excessive accumulation of fat, primarily caused by an energy imbalance. The storage of excess energy in the form of triglycerides within the adipocyte leads to lipotoxicity and promotes the phenotypic switch in the M1/M2 macrophage. These changes induce the development of a chronic state of low-grade inflammation, subsequently generating obesity-related complications, commonly known as metabolic syndromes. Over the past decade, obesity has been studied in many animal models. However, due to its competitive aspects and unique characteristics, the use of zebrafish has begun to gain traction in experimental obesity research. To counteract obesity and its related comorbidities, several natural substances have been studied. One of those natural substances reported to have substantial biological effects on obesity are flavonoids. This review summarizes the results of studies that examined the effects of flavonoids on obesity and related diseases and the emergence of zebrafish as a model of diet-induced obesity.

## 1. Introduction

Obesity is a pathological condition characterized by an excessive accumulation of body fat that causes serious damage to health. It is a consequence of an alteration in energy balance in which energy intake exceeds energy expenditure, leading to an increase in body mass, of which 60–80% is usually fat [1]. Currently, across the globe, the number of overweight and obese people exceeds that of those suffering from under-nutrition [2] and approximately 11 million people die each year worldwide from the consequences of excess weight [3,4]. The most widely used indicator of obesity is BMI (Body Mass Index), the ratio of weight to height (weight/height^2^), according to which a subject is considered overweight for a BMI value greater than 25 or obese for a BMI greater than 30. This condition, which affect all social classes, is steadily increasing in all parts of the world and is, therefore, considered one of the major public health problems, widespread to epidemic proportions at the global level. It is considered a serious risk factor that facilitates the onset of chronic diseases, such as insulin resistance, diabetes, dyslipidemia, hyperglycemia, hypercholesterolemia, nonalcoholic fatty liver disease and nonalcoholic steatohepatitis, cancer and cardiovascular diseases (Figure 1) [5]. Obesity has a multifactorial origin, and any investigation of this epidemic must consider both exogenous and endogenous factors [5,6,7,8] and the interaction between biological, behavioral, genetic and environmental factors [9]. During the last decade, many strategies have been conducted to prevent and/or counteract obesity, including the use of natural compound. Flavonoids, the most studied group of polyphenols, have shown important biological effects on lipid, energy metabolism and weight status, in addition to their common function as free radical scavengers in in vivo and in vitro models. The possible reversal of obesity has been explored in many animal models. However, zebrafish is emerging as a promising model organism for obesity research with its various interesting aspects. For all the above-mentioned reasons, the present review aims to give an overview of flavonoids and their various beneficial effect on obesity and associated comorbidities in the obese zebrafish model.

## 2. Obesity: Model of Choice

Based on the analysis of the main causes that determine the increase in the prevalence of obesity, today the main challenge is the search for effective remedies [10]. In this regard, it is essential to validate new therapies and remedies to counter this pandemic. Thanks to the numerous studies on animal models, a lot of knowledge about the etiology of obesity has emerged. For this aim, it is necessary to choose a model animal organism to ensure the study of obesity and related disorders, the optimization of therapies to be translated to humans. Several model organisms are considered in the scientific world, and their selection in obesity studies requires the evaluation of many factors [11]. Initially, the choice of model organisms was based on similarities in size, body structure, and relevant physiology with humans. The canine model provides a valuable model for evaluating and examining metabolic effects related to overweight, obesity, and diabetes mellitus by allowing quantification of hepatic glucose uptake, which is impossible to assess directly in humans or small animal models [12,13]. Another mammalian model organism used for translational studies in obesity and diabetes mellitus research is the pig [14]. The remarkable similarity between neuro-anatomy, gastrointestinal tract, body composition, and omnivorous diet, the predisposition to the development of obesity, and related cardiovascular diseases, such as hypertension and atherosclerosis, are additional reasons to select this animal species for nutritional and pharmacological (preclinical) studies. As in humans, porcine obesity is induced by a high-energy or high-carbohydrate diet [14]. Among mammalian animal models, another of relevant interest is characterized by non-human primates which have close genetic homology and high physiological similarity to humans. Despite the similar physiology of mammalian animal models to humans, some limits compromise their choice as model organisms. They are undesirable for large-scale, time-efficient experiments because of their limited numbers, high maintenance costs, long life cycle and their low number of offspring [11]. In recent decades, one of the most widely used preclinical models to study metabolic disorders is represented by small rodents, including rats and especially mice [15]. These animal organisms are widely used as a Diet Induced Obesity (DIO) model to evaluate the interaction between diet and genes in obesity and insulin resistance. DIO involves the administration of calorie-rich foods, essentially a high-fat diet HFD or high in fructose diet or both, to monitor progression of obesity, diabetes, as well as related comorbidities [16]. Although the DIO mouse model helps mimic the pathogenesis of human obesity and allows for the superimposable evaluation of the effects of drugs or genetic manipulations on the development of obesity, it also has disfavor from those mentioned above for other mammalian models, essentially the moderately long lifecycle, and cost [17,18]. In addition, transgenic rodent lines have been widely used in experimental obesity studies. Numerous molecular genetic tools can engineer targeted or untargeted mutations, from single nucleotide exchanges to chromosomal rearrangements for gene function assays in mice [19,20]. Non-mammalian experimental model organisms are often used in early research as screening tools or to monitor body fat storage and function. The aforementioned models, and in particular metazoas, have advantages such as conserved insulin-like signaling and secretion, short life span, accelerating the quantification of the long-term and transgenerational consequences of obesity and diabetes. As in mammals, diet-induced obesity is also obtained in nematode Caenorhabditis elegans, the fruit fly Drosophila melanogaster and in zebrafish (*Danio rerio*) too [21,22,23,24,25,26,27].

### A Promising Model Organism for Obesity Research: Zebrafish

The zebrafish (*Danio rerio*) is an attractive experimental model that can offer new opportunities in the research fields due to a combination of interesting aspects and peculiar characteristics [28,29,30,31]. The choice of zebrafish as a rather advantageous model over others is of considerable relevance because it is small robust fish, easy to maintain, and economical to breed [32,33]. It can spawn throughout the year under laboratory conditions every 2–3 days and single brood may contain several hundred eggs. The development process is rapid (from fertile egg to adult it takes around 3–4 months) which makes it suitable for selection experiments. Additionally, compared to other fish, zebrafish eggs are larger with a diameter of 0,7 nm at fertilization. One of the aspects of remarkable interest is the optical transparency, in fact zebrafish are optically transparent from the moment of external fertilization until organogenesis. This allows for the easy in vivo observation of development and physiological processes in first weeks of life [28,34]. For these reasons, zebrafish can be used to trace the origin of adipogenesis, and screening defects in lipid uptake, transport and arrangement of lipids from egg fertilization to the larval state. Several dyes are used to visualize adipocytes during zebrafish larvae development, such as Sudanophilus dye, Oil Red O (ORO), Sudan, Nile Red or Lipid Green [35]. However, to assess fatty acid mobilization and its transportation, the lipophilic fluorophore BODIPY^®^ can be used [36]. Zebrafish are useful model organisms for in vivo testing, dedicated to preclinical tests; indeed, this approach allows the screening of large libraries of natural compounds including flavonoids for the treatment of obesity [37,38]. Drug screening is directly conducted on zebrafish larvae since they have the signal transduction pathways regulating lipid metabolism [39]. Additionally, considering the high genetic and physiological likeness with mammals, zebrafish are a useful agent for high-throughput screening, using a multiwell plate [40]. In addition to automated systems for handling/sorting, high-resolution image acquisition and data collection tools have significantly increased throughput screening [41,42,43].

The zebrafish is an excellent model for studying obesity because it has major organs that are important for regulating energy metabolism in mammals, including digestive organs, adipose tissue and skeletal muscle, in addition to the conservation of ortholog, appetite, insulin regulation and lipid storage, genes [32,44,45,46].

The overlap of zebrafish with mammalian adipose tissue distribution makes this teleost an excellent experimental model of obesity. At the larval stage, 8 days post fertilization (dpf), the beginning of adipogenesis in the visceral cavity clearly appears. From 12 dpf, the accumulation of adipocytes in the pancreatic area can be observed. At this stage, there is a correlation between the number of adipocytes and larval size, rather than the age [35], whereas, at 17 dpf, larvae show a correlation between visceral white adipose tissue development and age [47]. In adult zebrafish, the largest deposits of adipocytes are found in the visceral and subcutaneous regions, in pancreas and liver (Figure 2) but smaller deposits are found in the tail, mandible and periorbital regions at subcutaneous level, which provides the opportunity to understand the regulation of body fat distribution in obesity [47,48]. During the early phase of development, adipocytes may contain a single large lipid droplet or may contain multiple smaller lipid droplets, ranging in sizes from 1 to 100 μm [35]. All of these features are comparable to mammalian adipocyte development and demonstrate that zebrafish contain mammalian-like white adipose tissue [49]. Furthermore, homology in lipid metabolism between zebrafish, rats, mice, and humans has been demonstrated through comparative transcriptome analysis [35]. Several studies demonstrate that lipid metabolism pathways are conserved between fish and mammals [48,50,51,52,53], and the upregulation of microsomal triglyceride transfer protein (MTP), in response to a single high-fat meal, has been demonstrated in the proximal intestine and liver [51,54,55]. MTP induces the assembly of lipoproteins in the nascent apolipoprotein B (ApoB). The ApoB-MTP lipoprotein complex plays a preventive role in lipids degradation by proteasomes and increase plasma lipid levels [56,57]. The zebrafish MTP gene encodes a protein with 54% identity with human MTP [58]. Through in situ hybridization and RT-PCR, the team confirmed the developmental regulation and tissue specificity of MTP expression during early embryogenesis and in the anterior intestine and liver from 48 h post fertilization forward. A primary aspect of obesity is hypertrophy and hyperplasia of the adipocytes [35]. Moreover, in zebrafish, lipids are also stored in visceral, intramuscular and subcutaneous adipocytes [48]. The Artemia saline diet-induced obese (DIO) zebrafish model was first mentioned in 2010 by the Oka research group and, subsequently, it was reproduced by other research groups. The aforementioned model has also been used to test the effects of some dietary supplement effects on body fat accumulation [27,59,60,61,62,63]. DIO zebrafish have subsequently been documented and applied as an induction by overfeeding with commercial feed or through a high calorie diet of corn oil or commercial oil enriched feeds for various obesity study purposes [27,64,65,66]. Although it should be noted that not all lipid metabolism genes are highly conserved in sequence and function in zebrafish. Zebrafish Leptin, the adipostatic hormone that regulates fat mass and whose absence can causes hyperphagia and obesity, is only 19% identical to human protein [67,68]. Thanks to its fully sequenced genome, nowadays, the studies of obesity and related diseases, take advantage of the stable zebrafish obese transgenic line [33].

## 3. Polyphenols Classification and Flavonoid

The polyphenols, secondary metabolites of plants and fungi, representing a heterogeneous group of diverse phytochemicals characterized by a phenolic ring, which can be classified according to their chemical structures and biosynthetic pathways into four main groups (Figure 3): phenolic acids, flavonoids (with six or eight subgroups), stilbenoids, and lignans [69]. Among polyphenols, flavonoids are integrated as the first and largest subgroup of chemical compounds that constitutes plant and flower pigments, distributed in plant leaves, seeds, and flowers, and that shares the common function of being free radical scavengers by donating phenolic hydrogen atoms [69,70].

Flavonoids are low-molecular weight heterocyclic compounds synthesized from the products of the shikimate pathway. Structurally, they share a C6-C3-C6 skeleton of a two benzene rings (A and B) joined by a heterocyclic pyrone (C) and are classified according to their structure into several groups (Figure 4). Many flavonoids are found bound to sugars (glycosides), less frequently without a sugar group (aglycones) [71].

Flavonoid biosynthesis in plants is a process of acclimation to biotic (protection against pathogens, fungal parasites, herbivores) and abiotic stressors (protection against ultraviolet, radiation, temperature, day length) that explains, in the same species, the difference in flavonoid content [72]. The shikimate pathway produces aromatic amino acids, including phenylalanine, which can be further modified through the sequential actions of phenylalanine ammonia lyase (PAL), cinnamate 4 hydroxylase (C4H), and p-coumaroyl CoA ligase (PCL) to form p-coumaroyl CoA [73]. Under the action of the enzyme chalcone synthase (CHS), the condensation of one molecule of p-coumaroyl-CoA (future B ring of flavonoid) with three molecules of malonyl-CoA (future A ring) give chalcone (2′, 4′, 6′, 4-tetrahydroxychalcone). Chalcone is subsequently isomerized by the enzyme chalcone flavanone isomerase (CHI) to flavanone. From these central intermediates, the pathway diverges into several classes of flavonoids which are flavones (e.g., apigenin, and luteolin), flavonols (e.g., quercetin, and kaempferol), flavanones (e.g., naringenin, and hesperetin), isoflavones (e.g., genistein and daidzein), flavanols (e.g., catechin and epicatechin), and anthocyanidins (e.g., pelargonidin, cyanidin, malvidin, and delphinidin) (Figure 4) [72,74].

The bioavailability of flavonoids varies among different subclasses and has been shown to be quite poor in many of them, which has often hindered their pharmaceutical potential [75]. Multiple factors, including their chemical structure and solubility, interaction with endogenous enzymes that modulate certain pathways and the gut microbiota, play a crucial role in determining the absorption and bioavailability of flavonoids [76,77]. In the gastrointestinal tract, dietary flavonoids are enzymatically hydrolyzed by bacterial enzymes to release the aglycones, which are well absorbed in the small intestine compared to glycosides due to better membrane interactions [76,77,78,79]. The presence of sugars reduces the cellular absorption of glycosides, limiting the activity of gut microbiota and thus affecting the bioavailability of sugar conjugated flavonoids in vivo [76,77,80]. The type of glycosylation influences flavonoid hydrolysis in the intestine. Hydrolysis of C-glycosyl flavonoids, for example, is less efficient than that of O-glycosides. In contrast to the hydroxylated forms, O-methylated flavonoids possess better bioavailability due to their delayed absorption and increased permeability across membranes [77]. Unabsorbed flavonoids are metabolized by the tract microbiota to give smaller assimilable compounds. However, they may interact with glycoprotein P or permeability-glycoprotein, which is widely expressed throughout the gastrointestinal tract, thus preventing them from reaching the circulatory system [79,81].

In the last decade, many studies have shown that flavonoids exhibit a broad spectrum of biological properties and widespread beneficial effects, including antioxidant [82], antibacterial, anti-inflammatory, antithrombotic, cardioprotective, hepatoprotective, neuroprotective properties and their possible role in the prevention and treatment of various chronic diseases, such as hypertension, neurodegenerative diseases, and an inverse relationship has been proposed between flavonoid intake and the risk of obesity and its associated metabolic disorders [70,77]. This wild spectrum of biological and pharmacological properties is conferred by the diversity of the chemical skeleton of the flavonoid, which are often found to be hydroxylated, glycosylated or methoxylated at different positions [77].

## 4. Flavonoids, Obesity and Metabolic Syndrome

Obesity is a medical condition, defined by WHO as an abnormal or excessive accumulation of fat that can compromise health [4]. Energy imbalance in which energy intake is greater than energy expenditure is the primary cause of visceral or central obesity, as excess energy is stored as triglycerides within adipocytes, which increase in size (hypertrophy phynotype), and number (hyperplasic phenotype), or both [83,84]. By accumulating excess energy, adipocytes become hypertrophic, which causes the release of free fatty acids into the circulation (lipotoxicity), adipocytes change their immunological balance, which promotes, with adipose tissue resident immunes cells (macrophages), the production and the circulating levels of proinflammatory cytokines and decreases the concentration of anti-inflammatory adipokines, such as adiponectin [85,86,87,88]. These changes in adipose tissue lead to the development of chronic state of low-grade inflammation that secondarily generates obesity-related complications, commonly known as metabolic syndrome. This syndrome includes insulin resistance, hyperglycemia type 2 diabetes mellitus (T2DM), cardiovascular diseases, dyslipidemia (decreased concentration of cholesterol and triglycerides), steatosis, fibrosis, hypertension, heart attack [84,87]. The literature strongly suggests that flavonoids demonstrate an important biological effect on obesity, as demonstrated by their ability to lower fat mass, lipid droplets in the liver, and total triglycerides/cholesterol in both in vitro and in vivo models.

### 4.1. Effects of Flavonoids on Lipid Accumulation

Lipids accumulate is caused by an increase in the size of adipocyte (hypertrophy) and an increase in their number (hyperplasia). Hyperplasia is regulated by the differentiation of multipotent mesenchymal stem cells into preadipocytes that, under appropriate stimulation, can differentiate terminally into mature adipocytes. These adipocytes are capable to storing excess energy as cytoplasmic neutral lipid droplets of different sizes which can exceed 100 micrometers [61,89]. In this regard, it has been reported that flavonoids have great potential. They deal with lipid accumulation through numerous mechanisms, including the inhibition of adipocyte differentiation, primarily caused by the reduced expression of important regulatory adipogenic transcription factors, decreased lipogenesis, and induction of adipocyte apoptosis [90]. Thus, quercetin, a plant flavonol found in a wide variety of vegetables and fruits, reduces the lipid accumulation through decreasing preadipocyte differentiation, lipogenesis, and induction of adipocyte apoptosis [91,92]. The inhibition of adipogenesis was regulated by the downregulation of central transcriptional regulators of adipogenesis (SREBP-1, C/EBPα, and PPARγ) and FAS, a key adipogenic enzyme. Quercetin increasing apoptosis involved mitogen-activated protein (MAP) kinases, specifically the decrease in extracellular signal-regulated kinases (ERKs). Thus, inhibition of the extracellular signal-regulated kinases enhances apoptosis [92]. Kaempferol, another flavonoid, negatively regulates adipogenesis by downregulating PPARγ, aP2, and SREBP-1C [93] and CCAAT-enhancer-binding protein alpha [94] in 3T3-L1 adipocytes. In zebrafish, kaempferol has anti-adipogenic properties that regulate early adipogenic factors (KLF5, KLF4, KLF2, and C/EBPβ) [93]. Upregulation of lipolysis regulatory enzymes (Adipose triglyceride lipase (Pnpla2)) is one of the anti-adipogenic mechanism of kaempferol [94]. In addition, baicalein, a type of flavonoid originating from *Scutellaria baicalensis*, has a significant capacity to decrease lipid accumulation in zebrafish in a dose-dependent manner [95]. In 3T3-L1 cells, baicalein inhibit lipid accumulation during adipogenesis by arresting cell cycle in the G0/G1 phase through cyclin downregulation, suppressing the mRNA expression of early adipogenic factors. The above-mentioned factor shortage leads to the downregulation of late adipogenic factors, negatively regulating the m-TOR signaling pathway, involved in lipid accumulation during adipogenesis and decreased p-p38 MAPK and pERK levels in adipocytes [95,96]. *Puerariae Lobatae* radix flavonoids and puerarin successfully limited lipid accumulation in the abdomens of zebrafish larvae in a dose dependent manner [38].

### 4.2. Effects of Flavonoid on Triglycerides

Triglycerides (TG) are important factors in fat accumulation during adipocyte differentiation. During lipogenesis, glycerol-3-phosphate acyltransferase converts glyceraldehyde-3-phosphate, a glucose metabolite, to lysophosphatidic acid (LPA); lysophosphatidic acid acyltransferase-θ (LPAATθ) converts LPA into phosphatidic acid (PA), a biosynthetic precursor of acylglycerols. PA is converted into diacylglycerol (DAG) by lipin1 and diglyceride acyltransferase-1 (DGAT1) catalyzing the conversion of DAG into TG [91]. Lipin1 is linked to low-density lipoprotein secretion and PPARγ expression. PPARγ expression is stimulated by m-TOR signaling, and m-TOR together with AKT (the upstream factor of m-TOR) stimulates TG synthesis. Available data suggest that flavonoid inhibited triglycerol levels in a dose-dependent manner [93]. The lipogenic factors involved in TG synthesis (LPAATθ, DGAT1 and lipin1) are dependent on adipogenesis. During adipogenesis, early adipogenic transcription factors, such as CCAAT/enhancer-binding protein-β(C/EBPβ), induce key adipogenic factors, such as CCAAT/enhancer-binding protein-α(C/EBPα) and peroxisome proliferator-activated receptor-γ(PPARγ). The expression of C/EBPα and PPARγ activates lipid synthetic proteins, including fatty acid-binding protein 4 (FABP4) and lipin1 [91]. Flavonoids have the capability to inhibit TG accumulation. This effect can be explained by the effect of flavonoids to reduce the levels of C/EBPβ, C/EBPα and PPAR γ protein in a dose-dependent manner, resulting in reduced levels of LPAATθ, DGAT1 and lipin1. Thus, quercetin inhibited TG accumulation by >40%, while curcumin, an antiadipogenic phytochemical, inhibited TG accumulation by ~25% [91]. Baicalein, kaempferol, and eriocitrin, flavonoids that have lately been reported, significantly suppressed the increase in TG in DIO-zebrafish [38,93,95].

### 4.3. Flavonoid and Cholesterol

The DIO zebrafish model, has a high cholesterol level. Many flavonoids, previously studied for anti-obesity effects, have the ability to reduce cholesterol levels. Puerariae Lobatae radix flavonoids and puerarin significantly downregulated the elevated mRNA levels of the 3-hydroxy-3-methylglutaryl coenzyme A reductase b (HMGCRB), the key enzyme involved in lipid metabolism and cholesterol biosynthesis, in zebrafish larvae [38]. On the other hand, baicalein decreased the expression levels of SREBP1 and fatty acid synthase (fasn) and regulated the synthesis and desaturation of fatty acids, proteins and a master gene regulating cholesterol synthesis. Naringenin instead significantly induced a downregulation of the zebrafish larva mRNA of the fatty acid desaturase 2 (fads2), a dyslipidemia-related gene which influences the concentrations of total cholesterol, low-density lipoprotein cholesterol, high-density lipoprotein cholesterol and triglycerides, fasn, enoyl-CoA hydratase, short chain 1 (echs1), Fatty acid-binding protein 10a (fabp10α), HMG coenzyme A reductase a (hmgcra) and hmgcrb [97].

### 4.4. Effects of Flavonoid on Inflammation

Obesity is a chronic state of low-grade inflammation. During the development of obesity, progressive hypertrophy of adipocytes promotes tissue hypoxia and macrophages infiltration, which induce the increased secretion of various proinflammatory mediators, such as tumor necrosis factor alpha (TNF-α), interleukin 6 (IL −6), plasminogen inhibitor 1 (PAI-1), C-reactive protein (CRP), and monocyte chemoattractant protein 1 (MCP-1), among others, besides the decrease in the concentration of anti-inflammatory adipokines, such as adiponectin, characterizing a chronic inflammation of low grade 6 [98].

Additionally, during obesity, adipose tissue produces a greater amount of reactive oxygen species (ROS) which causes oxidative stress. This stress in turn leads to the abnormal adipokines production (chronic low-grade inflammation), where it has been shown, for example, that adiponectin concentration is inversely related to the concentration of ROS [99].

There are multiple flavonoids that have been shown to be useful as anti-inflammatory agent in the low-grade inflammation that occurs in obesity. Quercetin, in particular, has been shown to have anti-inflammatory effects through the inhibition of MAPK signaling factors (ERK1/2, JNK and p38MAPK) and, consequently, to inhibit the secretion of inflammatory cytokines IL-1β and IL-6, MCP-1 and TNF-α in 3T3-L1 adipocytes and macrophages. In addition, quercetin has the ability to stimulate the amount of IL-10, a known inhibitory factor of cytokine synthesis and anti-inflammatory cytokine [91]. In studies conducted with Puerariae Lobatae radix flavonoids, puerarin, and Citrus sinensis flavonoids, they were observed to possess important anti-inflammatory activity, assured by the downregulation of inflammatory cytokine-related genes, such as IL-1β (interleukin-1β), IL-6 and TNFα (tumor necrosis factor-α) in a zebrafish model [38,100]. The flavonoid-rich extract of Withania somnifera leaf, essentially with kaempferol and agigenein, revealed the significant inhibition of TNFα in adult zebrafish [101].

In the DIO-zebrafish visceral adipose tissue, the ability of yuzu peel, vinaccia, and auraptene to upregulate the expression of adipokines, adiponectin negatively correlated with markers of inflammation and oxidative stress, has been demonstrated [102,103].

Medlar leaf, grape skin and acai puree are three major sources of flavonoids tested for their anti-atherosclerotic and anti-diabetic activity. These plant extracts have shown antioxidant, anti-inflammatory and anti-atherosclerotic activities [104].

### 4.5. Insulin Resistance

Insulin resistance (IR), defined as a decreased ability of cells to respond to insulin stimulation, is a crucial feature of prediabetes and is the first detectable abnormality in type 2 diabetes mellitus, a progressive metabolic disorder characterized by high blood glucose concentration, abnormalities in carbohydrate, lipid, and protein metabolism, [69,105,106]. In the early phase of IR, normal pancreatic β cells increase insulin production to compensate for IR and glucose utilization remains relatively normal. However, when IR continues, β cells gradually fail to secrete adequate amounts of insulin for metabolic compensation, leading to insulin insufficiency and impaired glucose tolerance. IR usually occurs in peripheral tissues, such as liver, adipose, and skeletal muscle [105,106]. Accumulating evidence shows that inflammation initiated by adipose tissue is a major contributor to the development of IR and T2D. Elevated levels of proinflammatory cytokines (tumor necrosis factor α (TNF-α), interleukin 6 (IL-6), interleukin 1β (IL-1β) and resistin) secreted by adipose tissue and macrophages infiltrating, as well as decreased levels of anti-inflammatory cytokines (interleukin 10 (IL-10) and adiponectin), have been reported in various diabetic and IR states [105,106]. The release of various cytokines and chemokines promotes the migration and activation of macrophages, which are recruited to the islet and further enhance the inflammatory environment by exacerbating the release of cytokines causing β-cell loss [106,107]. In the aforementioned section of our review, we extensively discussed the modulation of cytokines by flavonoids with an emphasis on therapeutic application against inflammation. Flavonoids, which showed anti-inflammatory properties, have been suggested as an excellent candidate to prevent hyperglycemia and the IR of zebrafish larvae by targeting inflammatory signals [69,91,104,106,108].

### 4.6. Flavonoid Effects on Non-Alcoholic Fatty Liver Disease

Non-alcoholic fatty liver disease (NAFLD), is an excessive accumulation of neutral lipids in the liver due to elevated hepatic lipogenesis, and low hepatic excretion of very low-density lipoprotein (VLDL) associated with metabolic syndrome, particularly obesity, insulin resistance, type 2 diabetes and cardiovascular disease and encompasses a spectrum of liver disorders, ranging from steatosis to non-alcoholic steatohepatitis (NASH), fibrosis, cirrhosis, and hepatocellular carcinoma [70,109,110,111]. In zebrafish, diet composition has a significant impact on the development of NAFLD. It is known that the excessive consumption of calories through overeating or a high fat or high sugar diet can induce obesity and hepatic steatosis [109,112].

Several flavonoids have been found to have positive effects on lipid metabolism, insulin resistance and inflammation, the most important pathophysiological pathways in NAFLD [70]. The effect of flavonoids on NAFLD was conducted through upregulation of PPARα which stimulates β-oxidation, mitigate inflammation and increases energy expenditure [70,102]. Another target identified in the treatment of NAFLD are fatty acid synthesis (FASN) associated genes and sterol regulatory element-binding protein 1 (SREBP-1C), a transcription factors that controls de novo lipogenesis through the induction of lipogenic enzymes that stimulate steatosis [70]. In zebrafish, several flavonoids, such as kaempferol and baicalein, reduce both SREBP-1c and FASN protein and gene expression [93,95]. Flavonoid-rich yuzu peel extract, increased mRNA expression of lipid oxidation markers (pparab, the zebrafish homolog of human *pparaγ*, and *acadm* in liver, *pparg* in adipose tissue, and *acox1* in both) causing lipid removal in the liver [70,102]. In addition, both flavonoid-rich extracts of medlar leaf, and acai puree, have anti-oxidant, anti-inflammatory activity and reduce hepatic steatosis in a hypercholesterolemic zebrafish model [104]. Phytochemical studies revealed that *Salvia plebeia* mainly contains flavonoids, such as apigenin, hispidulin, homoplantaginin, nepetin, nepetin-7-glucoside, and luteolin [113]. A study conducted in zebrafish, proves the effect of the ethanolic extracts of *S. plebeia* to reduce fat vacuoles, lipid accumulation and hepatic steatosis by reducing the expression of lipid metabolism genes [114]. Eriocitrin significantly suppresses the increase in plasma TGs in DIO-zebrafish, reduced lipid droplets in the liver tissues by activating mitochondrial functions (upregulation of cox4i1 and atp5j), ATP synthesis (upregulation of cox4i1 and atp5j) and by upregulating the mRNA level of lipid metabolism genes, *pparab, acox1* and *acadm* [110]. Naringenin is another flavonoid known to reduce hepatic lipid accumulation in zebrafish, as well as in other models [97,115,116].

## 5. Conclusions

Many processes implicated in obesity and metabolic syndrome can be regulated by naturally occurring bioactive compounds that represent an effective alternative for developing cost-effective, biological and anti-obesity substances. Many experiments in phenolic compounds with anti-obesity effects have been carried out, both in vitro and in vivo. In particular, in the last decade, the obese zebrafish has become a reliable model, easy to obtain, with a similar response to higher vertebrate models. The beneficial effects of the natural compounds’, particularly those of flavonoids, have been tested and confirmed in obese models in which the reduction in body fat and the counteraction of related comorbidity action have been verified. Future studies on the use of the zebrafish model for diet interaction are needed to elucidate the processes related to those nutrition-related disorders, hopefully translating this information to human health.

## Figures and Tables

**Figure 1 molecules-26-03014-f001:**
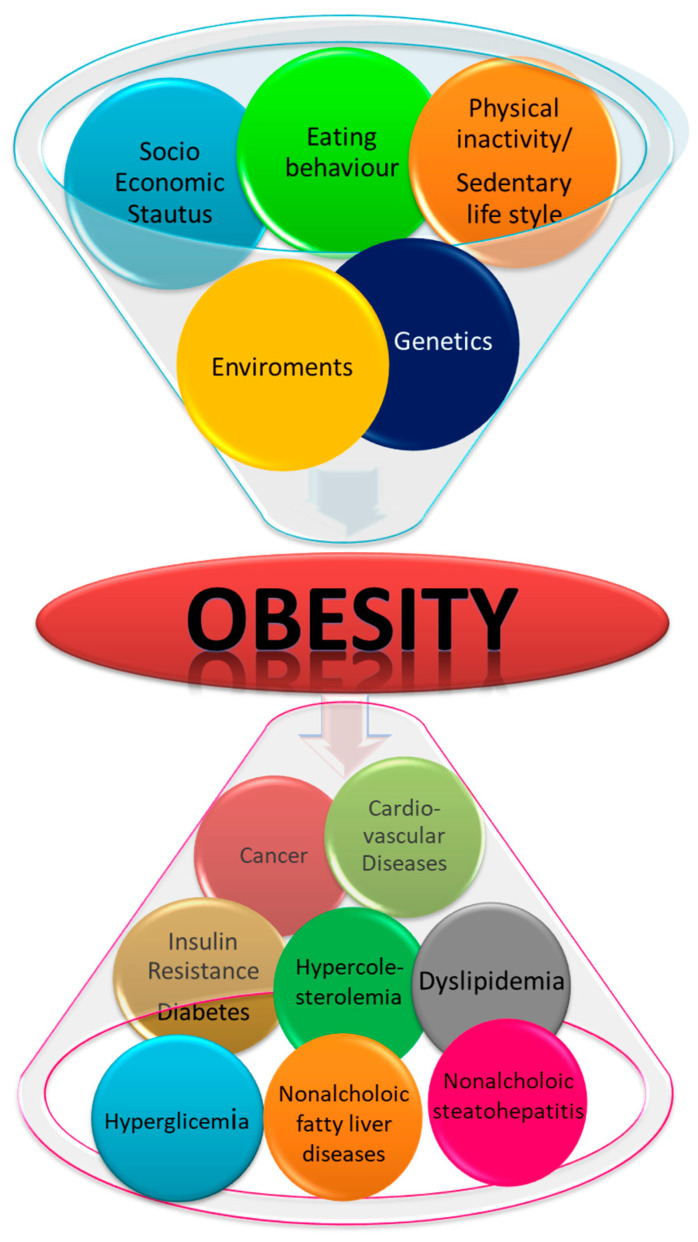
Scheme for causes of obesity and related disease.

**Figure 2 molecules-26-03014-f002:**
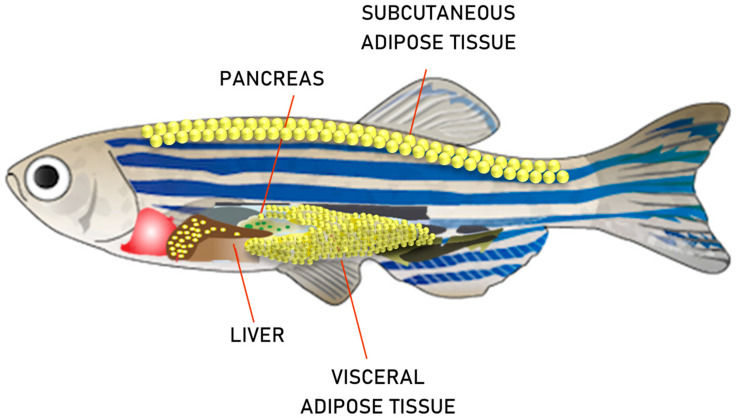
Principal adipose tissue depots in zebrafish.

**Figure 3 molecules-26-03014-f003:**
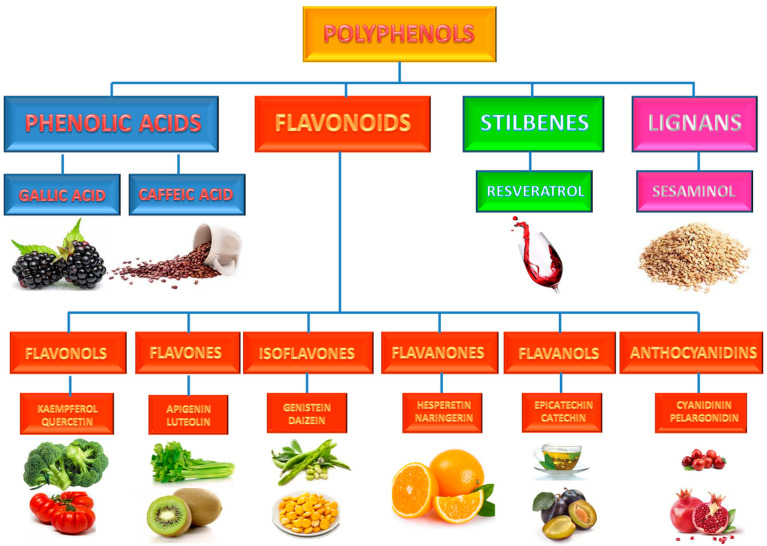
Classification of Pholiphenols and Flavonoids.

**Figure 4 molecules-26-03014-f004:**
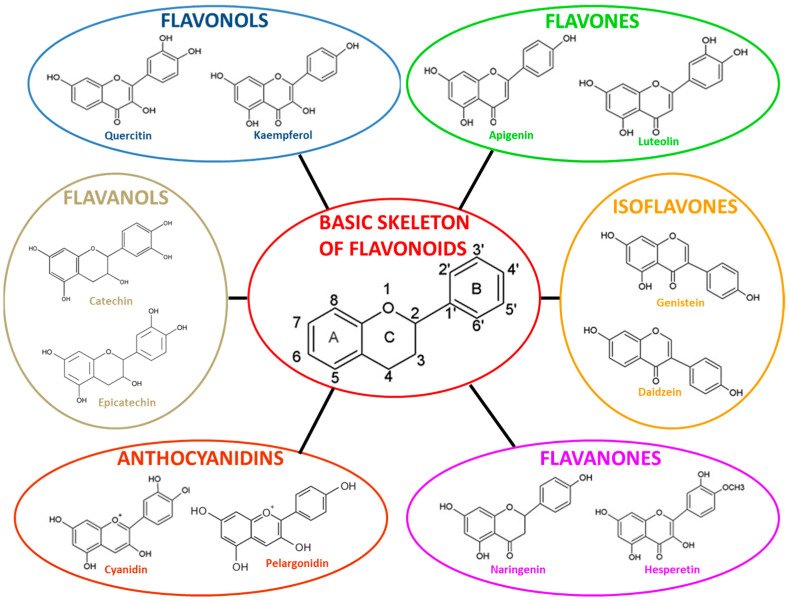
Chemical structures of flavonoids.

## Data Availability

Data sharing not applicable.

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
