# Peer review of "Zebrafish and Flavonoids: Adjuvants against Obesity"

_molecules, 2021, doi:10.3390/molecules26103014_

Round 1

Reviewer 1 Report

The aim of this study was to review the studies on the effects of flavonoids on obesity and related diseases and to evaluate the use of zebrafish as a model for diet-induced obesity. I think the topic of this work is interesting for the readers of Molecules journal. The manuscript is a good quality contribution but I suggest revising the paper in terms of language as there are several errors. I just indicated some below. I also suggest revising the title of the paper as it doesn’t reflect the content very well.

Some other comments are below:

Page 1, line 32:  “…indicators of obesity are the BMI…”

Page 1, line 37: “..chronic diseases such as insulin resistance..”

Page 1, Line 45, “… have shown an important biological…” remove “an”

Line 54: Caption of Figure 1 “Scheme for causes of obesity and related diseases.”

Page 2, line 62: “…translated to humans…”

Page 3, Line 97: “as a conserved insulin-like signaling” remove “a”

Page 4, line 123: “In addition, zebrafish has main functions well preserved” please revise this part, it is not clear.

Page 4, Line 124: Please revise “lipid storange”

Page 4, line 162: Please revise “now a day”

Page 5, line 167: I suggest revising as “Classification of Polyphenols and Flavonoids”

Page 6, line 186: “explain”

Page 6, Line 195:  Please revise “couting”

Page 6, Line 201: Caption of Figure 4. Chemical structures of flavonoids.

Under section 3, I think there is no need to have subtitles, as they are really short and can be combined as whole. Figure 4 may not be really necessary considering the aim of the work and also this is a very well known classification, but if the authors prefer to keep it, it is also fine.

Page 8, line 270: “Kaempferol,…”

Page 10, Line 369: Please revise this sentence for clarity “Elevated levels of proinflammatory cytokines,…”

Page 10, line 378: Please revise as “Flavonoids, which showed anti-inflammatory properties, have been suggested…”

Page 10, line 398:  Please revise as “In zebrafish, several flavonoids such as..”

Page 10, Line 399: Please write the names of flavonoids in small letters in the whole text as “kaempferol,  baicalein”

Page 10, Line 403: Revise as “shared anti-oxidant, anti-inflammatory activity and..”, and instead of “shared” another word would better be used.

Page 10, Line 405: “Salvia plebeia” Please write it in italic and check the whole text for such cases.

Page 10, Line 407: “A study conducted in zebrafish, proved…” Please use past tense and check the whole manuscript for language.

I think the authors should cite their recently published paper on “Zebrafish as a Useful Model to Study Oxidative Stress-Linked Disorders: Focus on Flavonoids”

Author Response

Dear Reviewer,

We would like to thank you for taking the time and effort necessary to review the manuscript. We sincerely appreciate all valuable comments, which helped us to improve the quality of the article. Our responses to your comment are described below in a point-to-point manner.

The aim of this study was to review the studies on the effects of flavonoids on obesity and related diseases and to evaluate the use of zebrafish as a model for diet-induced obesity. I think the topic of this work is interesting for the readers of Molecules journal. The manuscript is a good quality contribution but I suggest revising the paper in terms of language as there are several errors.

The language has been revised by native speaker.

I just indicated some below. I also suggest revising the title of the paper as it doesn’t reflect the content very well.

The title has been revised.

Some other comments are below:

Page 1, line 32:  “…indicators of obesity are the BMI…”

Has been correct (line 32).

Page 1, line 37: “..chronic diseases such as insulin resistance..”

Has been correct (line 38).

Page 1, Line 45, “… have shown an important biological…” remove “an”

Has been correct (line 45).

Line 54: Caption of Figure 1 “Scheme for causes of obesity and related diseases.”

Has been correct (line 54).

Page 2, line 62: “…translated to humans…”

Has been correct (line 62).

Page 3, Line 97: “as a conserved insulin-like signaling” remove “a”

Has been correct (line 97).

Page 4, line 123: “In addition, zebrafish has main functions well preserved” please revise this part, it is not clear.

The sentence has been rewrite (lines 128-131).

Page 4, Line 124: Please revise “lipid storange”

Has been corrected (line 131).

Page 4, line 162: Please revise “now a day”

Has been revised (line 170).

Page 5, line 167: I suggest revising as “Classification of Polyphenols and Flavonoids”

Has been revised (line 184).

Page 6, line 186: “explain”

Has been correct (line 192).

Page 6, Line 195:  Please revise “couting”

Has been revised (line 201).

Page 6, Line 201: Caption of Figure 4. Chemical structures of flavonoids.

Has been revised (line 207).

Under section 3, I think there is no need to have subtitles, as they are really short and can be combined as whole.

We have removed the subtitles in the section 3 as you have suggested.

Figure 4 may not be really necessary considering the aim of the work and also this is a very well known classification, but if the authors prefer to keep it, it is also fine.

Thank you for your suggestion. We prefer to keep the classification.

Page 8, line 270: “Kaempferol,…”

Has been corrected (line 274).

Page 10, Line 369: Please revise this sentence for clarity “Elevated levels of proinflammatory cytokines,…” The sentence has been revised (lines 370-374).

Page 10, line 378: Please revise as “Flavonoids, which showed anti-inflammatory properties, have been suggested…”

The sentence has been revised (line 380).

Page 10, line 398:  Please revise as “In zebrafish, several flavonoids such as..”

The sentence has been revised (line 400).

Page 10, Line 399: Please write the names of flavonoids in small letters in the whole text as “kaempferol,  baicalein”

The name of flavonoids has been changed in small letters all over the text.

Page 10, Line 403: Revise as “shared anti-oxidant, anti-inflammatory activity and..”, and instead of “shared” another word would better be used.

Shared has been changed (line 405).

Page 10, Line 405: “Salvia plebeia” Please write it in italic and check the whole text for such cases.

In the whole text Salvia plebeian has been rewrite in italic.

Page 10, Line 407: “A study conducted in zebrafish, proved…” Please use past tense and check the whole manuscript for language.

Was added the past tense (line 409)Grammar has been revised  by a native speaker through the whole manuscript.

I think the authors should cite their recently published paper on “Zebrafish as a Useful Model to Study Oxidative Stress-Linked Disorders: Focus on Flavonoids”

The recent paper has been  added (227).

Reviewer 2 Report

Introduction is well written with recommendation to accentuate obesity as the real pandemic of the 21st century. Line 29: replace word with world. Line 47: put phrases “in vitro” and “in vivo” into Italic font style (as well as through whole text).

Lines 60-62: double check English language. Lines 70-101: reformulate text to be more precise and correct references style. Line 127: abbreviation “dpf” is appeared for the first time, please define it. Line 143: put abbreviation “mtp” in uppercase. Lines 152-156 and 243-248: reformulate long sentences with two shorter ones. Line 168: article “the” should be used for plural. Lines 270 and 293: use Sentence cases. Line 311: change abbreviation “DOI” with “DIO”. Lines 349 and 399: use lowercase. Line 376: word afromentionned is incorrect. Line 422: use saxon genitive for plural (natural compounds beneficial).

Author Response

Dear Reviewer,

We would like to thank you for taking the time and effort necessary to review the manuscript. We sincerely appreciate all valuable comments, which helped us to improve the quality of the article. Our responses to your comment are described below in a point-to-point manner.

Introduction is well written with recommendation to accentuate obesity as the real pandemic of the 21st century.

Line 29: replace word with world.

Has been corrected (line 29).

Line 47: put phrases “in vitro” and “in vivo” into Italic font style (as well as through whole text).

Has been corrected through the whole text.

Lines 60-62: double check English language.

English language has been revised.

Lines 70-101: reformulate text to be more precise and correct references style.

The text has been reformulated as well as the references style (68-100).

Line 127: abbreviation “dpf” is appeared for the first time, please define it.

Dpf has been defined (line 134).

Line 143: put abbreviation “mtp” in uppercase.

Done through the whole text.

Lines 152-156 and 243-248: reformulate long sentences with two shorter ones.

The two sentences have been divided into several sentences.

Line 168: article “the” should be used for plural.

Has been changed.

Lines 270 and 293: use Sentence cases.

The sentences cases has been used (lines 269-288).

Line 311: change abbreviation “DOI” with “DIO”.

Has been changed (line 314).

Lines 349 and 399: use lowercase.

Lowercase have been used.

Line 376: word afromentionned is incorrect.

Aforementioned has been corrected (line 377).

Line 422: use saxon genitive for plural (natural compounds beneficial).

Saxon genitive has been used (line 425).

Reviewer 3 Report

The review by Montalbano and colleagues addresses the use of the zebrafish as a model organism for studies on obesity and specifically how flavonoids can be assayed using the fish as a discovery tool that is a viable alternative to other animals. The article does an extensive revision of the literature on the physiology of obesity and related diseases and on how flavonoids are potential molecules that can have positive outcomes on disease and well-being.

This review article is comprehensive and well researched and it should be of interest to the readers of Molecules. Nonetheless, there are few points that should be addressed before acceptance of the paper.

The major flaw in the article is the lack of a vision on how the zebrafish can be of practical use in obesity research and for evaluating the effects of flavonoids in this context. A good case is made for the model by pointing out its physiological, anatomical and molecular similarities and homologies with mammalian models. However, what is it that makes the zebrafish better suited for these studies than mice or swine? Is it size, number of individuals that can be tested, genetic tractability, transparency, or cost? It should be clearly indicated which type of experiments the authors envision that would generate new findings and that would be difficult or impossible with other models. Are they genetic screens, anatomical surveys or in vivo access to tissues using advanced microscopy?

In section 2.1., the authors make a good case for the use of zebrafish for these studies but fail to mention what is –in the opinion of this reviewer- the most advantageous feature of the model: the fact that drug or compound screens can be carried out in high-throughput fashion given the availability of hundreds to thousands of individual animals when larvae are used, and the fact that individual larvae in a multiwell plate array can be assayed in very small volumes (using a small amount of the compounds). The potential for automation of a screen, combining robotics (for plate or sample handling) and efficient assays for evaluating output (colorimetric assays, GFP based screens or physiological/behavioral responses that are easy to quantify automatically), is the feature of zebrafish that can place it ahead of other models.

Finally, while the manuscript is perfectly understandable, there are numerous flaws in the use of English throughout the text which need attention. These are spelling, grammar and syntax errors that would benefit from proofreading by a native speaker.

Minor point.

The color selection and light text make it difficult to read the words within the circles in Figure 1. The authors should consider changing colors or contrasting the text with the background to make it easier to read.

Author Response

Dear Reviewer,

We would like to thank you for taking the time and effort necessary to review the manuscript. We sincerely appreciate all valuable comments, which helped us to improve the quality of the article. Our responses to your comment are described below in a point-to-point manner.

The review by Montalbano and colleagues addresses the use of the zebrafish as a model organism for studies on obesity and specifically how flavonoids can be assayed using the fish as a discovery tool that is a viable alternative to other animals. The article does an extensive revision of the literature on the physiology of obesity and related diseases and on how flavonoids are potential molecules that can have positive outcomes on disease and well-being.

This review article is comprehensive and well researched and it should be of interest to the readers of Molecules. Nonetheless, there are few points that should be addressed before acceptance of the paper.

The major flaw in the article is the lack of a vision on how the zebrafish can be of practical use in obesity research and for evaluating the effects of flavonoids in this context. A good case is made for the model by pointing out its physiological, anatomical and molecular similarities and homologies with mammalian models. However, what is it that makes the zebrafish better suited for these studies than mice or swine? Is it size, number of individuals that can be tested, genetic tractability, transparency, or cost? It should be clearly indicated which type of experiments the authors envision that would generate new findings and that would be difficult or impossible with other models. Are they genetic screens, anatomical surveys or in vivo access to tissues using advanced microscopy?

In section 2.1., the authors make a good case for the use of zebrafish for these studies but fail to mention what is –in the opinion of this reviewer- the most advantageous feature of the model: the fact that drug or compound screens can be carried out in high-throughput fashion given the availability of hundreds to thousands of individual animals when larvae are used, and the fact that individual larvae in a multiwell plate array can be assayed in very small volumes (using a small amount of the compounds). The potential for automation of a screen, combining robotics (for plate or sample handling) and efficient assays for evaluating output (colorimetric assays, GFP based screens or physiological/behavioral responses that are easy to quantify automatically), is the feature of zebrafish that can place it ahead of other models.

In the section 2.1 the practical use of zebrafish as an adapted vector in the screening of natural compounds, especially flavonoids, in the context of obesity has been added in the manuscript. We have pointed out the crucial aspects that make zebrafish a suited model for this kind of study, from which its small size, enabling the use of a massive number of subjects in a simple multiwell plate. We have pointed out as well the importance of the automated system of handling, acquisition, and collection of data in increasing the screening throughput (lines 119-127).

Finally, while the manuscript is perfectly understandable, there are numerous flaws in the use of English throughout the text which need attention. These are spelling, grammar and syntax errors that would benefit from proofreading by a native speaker.

Through whole manuscript has been revised by native speaker.

Minor point.

The color selection and light text make it difficult to read the words within the circles in Figure 1. The authors should consider changing colors or contrasting the text with the background to make it easier to read.

The words in figure 1 have been written in black.

Reviewer 4 Report

It is recommended to include articles from 2021 of zabrafish and obesity. Minor errors in the lines 137, 481, 487, 517,519, 587.

Author Response

It is recommended to include articles from 2021 of zabrafish and obesity.

Articles from 2021 of zebrafish and obesity have been added.

Minor errors in the lines 137, 481, 487, 517,519, 587.

The error has been corrected